Disc-shaped fossils resembling porpitids or eldonids from the early Cambrian (Series 2: Stage 4) of western USA

Lieberman Bruce S. blieber@ku.edu 1 2
Kurkewicz Richard 3
Shinogle Heather 4
Kimmig Julien 2
MacGabhann Breandán Anraoi 5
1 Department of Ecology & Evolutionary Biology, University of Kansas , Lawrence , KS , United States of America
2 Biodiversity Institute, University of Kansas , Lawrence , KS , United States of America
3 Pangaea Fossils , San Francisco , CA , United States of America
4 Microscopy and Analytical Imaging Laboratory, University of Kansas , Lawrence , KS , United States of America
5 Department of Geography, Edge Hill University , Ormskirk , United Kingdom
De Baets Kenneth
Electronic publication date: 2017 Jun 6
Publication date: 2017
Volume: 5
Electronic Location ID: e3312
Received 2017 Jan 6; Accepted 2017 Apr 13
Copyright: ©2017 Lieberman et al.
Copyright year: 2017
Copyright holder: Lieberman et al.
License: This is an open access article distributed under the terms of the Creative Commons Attribution License, which permits unrestricted use, distribution, reproduction and adaptation in any medium and for any purpose provided that it is properly attributed. For attribution, the original author(s), title, publication source (PeerJ) and either DOI or URL of the article must be cited.
License URL: https://creativecommons.org/licenses/by/4.0/

Keywords: Porpitid, Hydrozoa, Cnidaria, Cambrian, Burgess shale type fossil, Elemental mapping, Eldonid

Funding: National Science Foundation EF-1206757 This research was supported by the National Science Foundation (EF-1206757). The funders had no role in study design, data collection and analysis, decision to publish, or preparation of the manuscript.

==============================
The morphology and affinities of newly discovered disc-shaped, soft-bodied fossils from the early Cambrian (Series 2: Stage 4, Dyeran) Carrara Formation are discussed. These specimens show some similarity to the Ordovician Discophyllum Hall, 1847; traditionally this taxon had been treated as a fossil porpitid. However, recently it has instead been referred to as another clade, the eldonids, which includes the enigmatic Eldonia Walcott, 1911 that was originally described from the Cambrian Burgess Shale. The status of various Proterozoic and Phanerozoic taxa previously referred to porpitids and eldonids is also briefly considered. To help ascertain that the specimens were not dubio- or pseudofossils, elemental mapping using energy dispersive X-ray spectroscopy (EDS) was conducted. This, in conjunction with the morphology of the specimens, indicated that the fossils were not hematite, iron sulfide, pyrolusite, or other abiologic mineral precipitates. Instead, their status as biologic structures and thus actual fossils is supported. Enrichment in the element carbon, and also possibly to some extent the elements magnesium and iron, seems to be playing some role in the preservation process.

Introduction

Aspects of the Phanerozoic fossil record of disc-shaped fossils in general, and jellyfish (medusozoans) fossils in particular, are somewhat cryptic, as the amount of character information generally preserved with such soft-bodied cnidarian specimens tends to be limited (though see Ossian, 1973; Cartwright et al., 2007; Liu et al., 2014 for exceptions); thus, any conclusions must be made with some caution (Hagadorn, Fedo & Waggoner, 2000). This is especially apposite given Caster’s (1942, p. 61) cautionary remark that “long scrutiny of problematical objects has been known to engender hallucination.” The degree of inscrutability increases when we extend our purview back to the Neoproterozoic, an interval from which many discoidal fossils exist (MacGabhann, 2007; MacGabhann, 2012; MacGabhann, 2014). Recently, MacGabhann (2007), MacGabhann (2012), MacGabhann (2014) and Young & Hagadorn (2010), Sappenfield, Tarhan & Droser (2016) provided a comprehensive overview of disc-shaped and medusoid fossils, such that detailed consideration of the phylogenetic affinities of a broad range of disc-shaped fossils and medusoids need not be undertaken herein. Instead, the focus here is on some new material recovered from the Echo Shale Member of the Carrara Formation (early Cambrian: Series 2, Stage 4, Dyeran) that seems to resemble fossil specimens at times treated as either porpitids or eldonids. As part of a discussion of the affinities of this new material, the fossil record of porpitids is also briefly considered.

Figure 1 Locality and stratigraphy.

(A) Map indicating where the specimens were derived from in the Nopah Range, Nevada, USA, with locality indicated by the star which represents 35°53′35.56″N 116°04′39.27″W; (B) A generalized stratigraphic chart for the Carrara Formation, with the star indicating the member the specimens were collected from.

Geology and paleoenvironment

The Carrara Formation is a regionally extensive, relatively shallow-water, mixed carbonate-siliciclastic unit of lower to middle Cambrian (Dyeran to Delamaran; Bonnia-Olenellus Biozone to Glossopleura Biozone) age in southern Nevada and southeastern California (Fig. 1A; Barnes & Palmer, 1961; Barnes, Christiansen & Byers, 1962; Palmer & Halley, 1979; Adams, 1995; Webster, 2011; Harwood Theisen & Sumner, 2016). It consists of mixed carbonate and siliciclastic sediments and varies in thickness between 300–500 m (Adams & Grotzinger, 1996; Keller, Lehnert & Cooper, 2012). Previous investigations indicate deposition in peritidal to shallow-subtidal conditions (Palmer & Halley, 1979; Keller, Lehnert & Cooper, 2012).

The Echo Shale Member was deposited in a lagoonal environment and is dominated by shales and siliceous mudstones, interbedded with silt- and sandstone beds; it is thickest in the Striped Hills area and thins out to the northwest (Palmer & Halley, 1979; Adams, 1995). It lies within the Bolbonelellus euryparia Biozone (Webster, 2011), overlays the Thimble Limestone Member, and in turn is overlain by the Gold Ace Limestone Member (Fig. 1B). The member is fossil poor and only a few trilobite species have been reported in the literature (Palmer & Halley, 1979).

The specimens were collected in the Nopah Range, California, USA, 35°53′35.56″N 116°04′39.27″W, at an elevation of about 820 m, and derived from float closely associated with greenish siliceous mudstones of the Echo Shale Member of the Carrara Formation. The rock slab the specimens were on also contained specimens of an olenelloid trilobite (KUMIP 431473), probably Bristolia (Harrington, 1956), confirming the stratigraphic assignment.

Materials and Methods

In any instance involving putative fossils of simple morphology that contain few diagnostic characters it is necessary to ascertain the biogenicity of the samples (Ruiz et al., 2004; MacGabhann, 2007; Kirkland et al., 2016). To help verify that the specimens were not abiological, pseudo- or dubiofossils sensu (Hofmann, 1971; Hofmann, Mountjoy & Teitz, 1991; Gehling, Narbonne & Anderson, 2000; MacGabhann, 2007), elemental mapping utilizing energy dispersive X-ray spectroscopy (EDS) was conducted using an Oxford Instruments 80 mm2 x-Max silicon drift detector (SDD), mounted on an FEI Versa 3D Dual Beam. The use of this approach applied to fossils in general, and Burgess Shale type fossils in particular, was pioneered by Orr, Briggs & Kearns (1998). It has also been employed to study Ediacaran fossils by Laflamme et al. (2011) and Cai et al. (2012), and MacGabhann (2012) has applied it to specimens of D. peltatum from a different locality. Analyses conducted in the present study used a horizontal field width of 2.39 mm, a kV of 10, a spot size of 4.5, and a 1,000 µm opening (no aperture). EDS maps were collected at a pixel resolution of 512 × 512 with a total of 18 passes. Analyses were conducted on two different parts of University of Kansas, Biodiversity Institute, Division of Invertebrate Paleontology (KUMIP) specimen 389538 (the best preserved specimen).

The specimens in Fig. 2 were photographed using a Canon EOS 5D Mark II digital SLR camera equipped with Canon 50 mm macro lens. The specimens in Fig. 3 were photographed using an Olympus UC50 camera attached to an Olympus SZX16 stereo microscope equipped with an Olympus SDF PLAPO 0.5XPF lens. Pictures were taken with specimens submerged in alcohol. The contrast, color, and brightness of images were adjusted using Adobe Photoshop.

Figure 2 The slab containing the fossil specimens.

(A) Part and (B) counterpart, where 1, KUMIP 389538; 2, KUMIP 389539; 3, KUMIP 389540. Scale bar is 10 mm.

Figure 3 cf. Discophyllum sp. Hall, 1847 from the Echo Shale Member of the Carrara Formation.

(A–D) Dorsal view of the part of KUMIP 389538. In (A) scale bar is 1 mm, the boxes surrounded in black represent locations of (C) and (D), and the boxes surrounded in blue were the regions subjected to EDS analysis with the results from these shown in Figs. 4 and 5 , respectively; (B) Line drawing illustrating the preserved structures; (C, D) Close-ups of different portions of the specimen; scale bars are 500 µm; (E) Dorsal view of the part of KUMIP 389540; scale bar is 1 mm; (F) Dorsal view of the part of KUMIP 389539; scale bar is 1 mm.

The biota of the Echo Shale Member consists of olenelloid trilobites, possible agnostoids, and the herein illustrated disc-shaped fossils. The disc-shaped fossils are preserved as part and counterpart of brown-grey carbonaceous films, and specimens KUMIP 389538 and KUMIP 389540 preserve some interior structure. The outer edge of KUMIP 389539 is vaguely preserved and the missing interior structure suggests partial decomposition of the type described by Kimmig & Pratt (2016). This could be due to scavenging (an unidentified phosphatic fossil is preserved next to it), pre-burial microbial decomposition, and/or diagenetic effects. The specimens are flattened, and that appears to have generated minor concentric wrinkles at the edge, best seen in KUMIP 389538. (MacGabhann (2012) provided a discussion of the taphonomy and preservation of Discophyllum specimens from the Ordovician of Morocco.) The Bristolia trilobite on the slab preserves the cephalon, and possibly part of the thorax, and appears to have been preserved completely articulated (Fig. 2A). The bulk of the thorax and pygidium are missing though because the specimen sits at the edge of the slab.

Results

Results derived from both EDS analyses are congruent (Figs. 4 and 5). The bulk mineralogy of the specimens was determined to be equivalent to that of the surrounding rock: either SiAlO or SiFeAlO depending on the part of the fossil/matrix analyzed. Spectral maps indicated the following variations in percentage by weight for different detectable elements: Si, 23.1–24.0%; Al, 13.7–14.2%; Fe, 7.0–16.8%; K, 4.2–6.3%; Ca, 1.1–2.0%; Na, <.1–1.1%; Mg, <.1–.8%; Mn <.1–.5%; Ti, <.1–.4%; P < .1–.2%; and S <.1–.1% (see included Supplemental Information). Given that Mn was barely detectable (.5%) or below detectable levels (<.1% in sample illustrated) in both the fossil and the surrounding matrix (see included Supplemental Information), the fossil cannot be the typically inorganic mineral precipitate pyrolusite. Si, S, Al, K, Na, and Ti levels were found to be identical in the fossils and the surrounding matrix (Figs. 4 and 5). Fe levels were primarily uniform throughout both the rock and fossil for the sample analyzed, although in one instance Fe levels are slightly elevated, both on and off of the specimen (Figs. 4 and 5). This, in conjunction with the fact that the sample morphology is not in line with typical, abiologic mineral precipitates, indicates that the fossils were not simply some form of inorganic mineral precipitate such as hematite, pyrite, or marcasite. Mg levels are primarily uniform throughout, although again there are a few elevated patches on and off the specimen (Figs. 4 and 5). There are only three elements that show any consistent elevation associated with the fossil (Figs. 4 and 5). The first is C, which seems to be elevated in moderately large, rounded patches, distributed seemingly at random across the fossils, and also along the margin of the specimen (Figs. 4 and 5). In a few cases C is slightly elevated, though in much lower densities in terms of both patch size and distribution, in the surrounding rock. The patchiness of the C may indicate partial weathering of the fossil. Ca is also elevated in places, with a few moderately large, rounded patches, but these are distributed only on parts of the fossils, and also along the margin of the fossil (Figs. 4 and 5). The Ca could perhaps represent recent diagenetic alteration associated with weathering or early diagenetic cement. Finally, P is uniformly distributed in the fossil and the surrounding matrix at low levels, except there appears to be some elevation along the margins of the specimen (Figs. 4 and 5); the preservation of these specimens does not appear to represent the type of phosphatization described by Xiao, Zhang & Knoll (1998).

Figure 4 Element maps of KUMIP 389538 and surrounding rock matrix.

The region demarcated by the blue box labeled “Fig. 4” in Fig. 3A was analyzed. Scale bars are 1 mm. Element map images were generated using Oxford Instruments AZtecEnergy EDS software. These images were migrated into Adobe Photoshop 2014.2.1 CC to create a single figure. No image manipulations were performed.

Figure 5 Element maps of a different portion of KUMIP 389538 and surrounding rock matrix.

The region demarcated by the blue box labeled “Fig. 5” in Fig. 3A was analyzed. Scale bars are 1mm. Element map images were generated using Oxford Instruments AZtecEnergy EDS software. These images were migrated into Adobe Photoshop 2014.2.1 CC to create a single figure. No image manipulations were performed.

EDS analyses thus seem to indicate the fossils are at least partly preserved as a kerogenized carbon film, which is consistent with a specific type of soft-bodied, Burgess Shale type preservation that has been identified (Butterfield, 1990; Moore & Lieberman, 2009). Not all Burgess Shale type fossils show such a preservational style (Orr, Briggs & Kearns, 1998; Gabbott et al., 2004). Often, these fossils are replicated as clay minerals, with parts of the fossils elevated in characteristic elements present in clay minerals such as K, Al, and Mg (Orr, Briggs & Kearns, 1998); at other times pyrite can play a significant role in replicating tissues (Gabbott et al., 2004). The existence of some partial elevation for both Mg and Fe in the specimen analyzed may also indicate a role for clay minerals and pyrite in the preservation process as well. Moore & Lieberman (2009) did previously identify instances in the Cambrian of Nevada, USA, from localities relatively stratigraphically and geographically close to the locality these specimens come from, when soft-bodied fossils were preserved as carbon films; they also identified instances from these nearby localities when fossils were preserved as clay minerals and/or pyrite. Other taphonomic processes associated with enrichment in the elements P and Ca could perhaps be playing some role in the preservation of these porpitid fossils. Notably, the EDS analyses of MacGabhann (2012) suggested that somewhat different taphonomic processes were associated with the preservation of Discophyllum specimens from the Ordovician of Morocco, especially involving no prominent role for C, although this is perhaps not unexpected given their different sedimentology and reconstructed paleoenvironments relative to what is known from the Cambrian Carrara Formation.

Taxonomy: The specimens are tentatively placed with Discophyllum Hall, 1847, a monospecific genus for D. peltatum ((Hall, 1847), p. 277, pl. LXXV, Fig. 3) (see also MacGabhann, 2012, figs. 4.68, 4.69), originally described from the Upper Ordovician (Mohawkian) Trenton group, near Troy, New York, USA (see MacGabhann, 2012, figs. 3.28–3.30 for illustrations of the locality). The specimens are referred to cf. Discophyllum sp. Hall, 1847, and greater justification for this taxonomic assignment is provided below. More information on D. peltatum is also provided below and in: Walcott (1898, p. 101, pl. XLVII, figs. 1, 2); Ruedemann (1916, p. 26, pl. XLVII, figs. 1, 2; 1934, p. 31, pl. 12, figs. 1, 2); Chapman (1926, p. 14); Caster (1942, p. 83); Zhu, Zhao & Chen (2002, p. 180) (where it is referred to as D. paltatum); Fryer & Stanley (2004, p. 1117); and comprehensively in MacGabhann (2012, p. 122, figs. 4.68–4.113, figs. 5.15–5.53).

If Discophyllum is a porpitid, as has been previously suggested, it would be classified as: Phylum Cnidaria Verrill, 1865; Class Hydrozoa Owen, 1843; Subclass Hydroidolina Collins, 2002; Order Anthoathecata Cornelius, 1992; Suborder Capitata Kuhn, 1913; Superfamily Porpitoidea Goldfuss, 1818; and Family Porpitidae Goldfuss, 1818. This follows the most up to date treatments available: Daly et al. (2007) and World Register of Marine Species (2015). However, MacGabhann (2012) and MacGabhann (2014) suggested an alternative placement for this taxon in an enigmatic group that was formerly largely Cambrian in age, the eldonids, including the eponymous Eldonia Walcott, 1911. The material presented here is not sufficiently well preserved to ascertain a higher-level taxonomic assignment. For additional discussion about higher-level taxonomic assignments of fossil porpitids see Fryer & Stanley (2004) and also MacGabhann (2012); for discussion on the early fossil record of Cnidaria see Van Iten et al. (2014).

Referred specimens: KUMIP 389538–389540.

Remarks: A total of three closely associated specimens from a small slab were collected; they are each preserved as both part and counterpart. All specimens are ovate in overall form, having a slightly elongated antero-posterior axis. The presumed dorsal side preserves a prominent set of rays or ridges that radiate from the central region. These could be akin to the radial flutes and folds of the float of modern and fossil porpitids (see Yochelson, 1984 and Fryer & Stanley, 2004 for discussion) but also might represent other structures seen in eldonids by MacGabhann (2012) and MacGabhann (2014). In cases it appears that some of the rays or ridges may split (Fig. 3). It is not possible to determine if this was caused by post-mortem decay or represents actual biology. If the latter, it would be congruent with what MacGabhann (2012) identified as secondary or tertiary ridges in eldonids. The details of the central region are sometimes obscured, but in KUMIP 389538 and 389540 (Figs. 2 and 3) there appears to be a small ovate structure from which the rays radiate. The margins of the disc show a faintly scalloped pattern. Concentric corrugations are absent. There is no evidence of a keel or sail as should be found in Velella Lamarck, 1801 (see Fryer & Stanley, 2004). Evidence of structures lateral of the radial ridges or fibers seems to be lacking, so there does not appear to be evidence of tentacles extending beyond the margin of the float. All specimens are preserved in low relief, and thus do not have cap-shaped relief, nor do they show evidence of deformation consistent with compression of an originally cap-shaped relief. There is no evidence of a coiled sac or dissepiments of the type identified by MacGabhann (2012), but this could be due to relatively poor preservation. The type specimens of D. peltatum Hall, 1847 were originally reposited in the Troy Lyceum (see Walcott, 1898) (the Troy Lyceum became today’s Rensselaer Polytechnic Institute) and are now at the Field Museum of Natural History (see MacGabhann, 2012). We have provided two alternative taxonomic assignments, and we concur with Conway Morris, Savoy & Harris (1991, p. 149–150) that “in the absence of diagnostic soft-parts, placement of certain discoidal fossils in” what are today known as the capitates (formerly the chondrophorines), can be challenging.

Discussion

Most discoidal unbiomineralized fossils of Paleozoic age have been compared or referred to as one of three groups: cnidarian medusae (Young & Hagadorn, 2010), the capitate hydrozoans (Fryer & Stanley, 2004) (previously referred to as chondrophorines), or the eldonids (MacGabhann, 2012). Comparisons are also made to discoidal specimens of Ediacaran age (e.g., Kirkland et al., 2016).

Comparison with discoidal taxa of Ediacaran age: The vast majority of described unbiomineralized discoidal fossils have been found in sedimentary rocks of Ediacaran age. The Carrara specimens bear little resemblance to any material known from the Ediacaran (MacGabhann, 2007). The most apparent distinction is taphonomic, with Ediacaran discoidal specimens generally preserved as positive hyporelief casts or negative epirelief molds on bedding surfaces (MacGabhann, 2014), fundamentally different from the preservation of the Carrara specimens as carbonaceous compressions. This does not preclude a comparison, as species can, of course, have specimens preserved in more than one taphonomic style (e.g., Zhu et al., 2008; MacGabhann, 2012). However, more importantly, there is little morphological data to suggest a link between these specimens and any of Ediacaran age.

Certain discoidal impressions of Ediacaran-aged taxa have at times been assigned to the Hydrozoa in general and the Porpitidae in particular (for additional information on such Ediacaran-aged specimens see Sprigg, 1947; Wade, 1972; Glaessner, 1979; Fedonkin, 1981; Stanley & Kanie, 1985; Sun, 1986). There are few similarities between these specimens and those described herein, except for the overall discoidal shape. For example, Eoporpita medusa Wade, 1972 consists of a small concentrically ornamented disc surrounded by radial structures, while Hiemalora Fedonkin, 1982 has a prominent and generally smooth central disc, with much wider radial structures that show prominent relief (Narbonne, 1994). Cyclomedusa davidi possesses radial striations, but these do not continue into the central circular zone (Sprigg, 1947; Sprigg, 1949). None of these resemble the material described herein, which lacks clear concentric structures.

Comparison is rendered difficult, however, by the taxonomic irregularities and complexity between and within Ediacaran discoidal genera and species (MacGabhann, 2007). Many specimens assigned to Cyclomedusa Sprigg, 1947 consist solely of concentric rings and lack radial features entirely. The same is true of species referred to Spriggia Southcott, 1958. It is also true of Kullingia delicata (Fedonkin, 1981), which occurs in both Ediacaran rocks and in Lower Cambrian strata in Newfoundland (Narbonne et al., 1991). Notably, Kullingia appears to be a trace fossil (scratch circle) that was produced by an anchored, tubular organism (Jensen et al., 2002; Sappenfield, Tarhan & Droser, 2016). Other Ediacaran discoidal forms are now known to be pseudofossils (e.g., Menon et al., 2016).

None of these Ediacaran specimens are still thought to represent hydrozoans (e.g., Zhang, Hua & Reitner, 2006; Cartwright et al., 2007; MacGabhann, 2007, and references therein). Young & Hagadorn (2010) reiterated this perspective when they noted that in many of these taxa the radial structures cannot be interpreted as radial canals. Indeed, the Ediacaran discoidal fossils have been recognized as benthic organisms, rather than pelagic forms, since Seilacher (1984).

In fact, most discoidal Ediacaran fossils are now thought to represent holdfasts of epibenthic stalked organisms, with the differences between specimens often due simply to taphonomic variation. For instance, Gehling, Narbonne & Anderson (2000) identified three major morphs of Aspidella Billings, 1872, which they suggested represent holdfast taphonomic variants (see also Tarhan et al., 2015, but see MacGabhann, 2007). The specimens described herein differ from the Aspidella ‘type’ morph by the lack of a prominent central slit, from the ‘flat’ morph by the lack of concentric rings, and from the ‘convex’ morph by the lack of a prominent central boss (Gehling, Narbonne & Anderson, 2000). Indeed, there is no prima facie reason to suggest a holdfast nature for these fossils, with no evidence for a benthic habit or stalk attachment (Gehling, Narbonne & Anderson, 2000; Sappenfield, Tarhan & Droser, 2016). For similar reasons, cf. Discophyllum sp. is also different from the Ediacaran-aged material that Hofmann (1971) and Hofmann, Mountjoy & Teitz (1991) classified and illustrated as “dubiofossils” of questionable biological affinities.

Comparison to cnidarian medusae: Cambrian cnidarian medusae have been described from several localities, including multiple sites in the United States (Hagadorn, Dott Jr & Damrow, 2002; Cartwright et al., 2007; Hagadorn & Belt, 2008; Lacelle, Hagadorn & Groulx, 2008; Young & Hagadorn, 2010; Hagadorn & Miller, 2011; Sappenfield, Tarhan & Droser, 2016). These are generally large, preserved as molds and casts, with convex sediment rings, and have quadripartite cracks. Clear criteria for the recognition of ancient medusae have been outlined by Young & Hagadorn (2010). Other bona fide medusae preserve considerably more anatomy than seen in the Carrara discs (e.g., Cartwright et al., 2007; Adler & Röper, 2012). As for the comparison to Ediacaran discoidal taxa, the fossils described herein resemble bona fide medusae only in terms of the overall discoidal shape, making such an affinity unlikely.

Comparisons with fossil capitates: cf. Discophyllum sp. also differs from what seem to be bona fide fossil capitates. For instance, it differs from the capitate Palaelophacmaea valentinei Waggoner & Collins, 1995 from the Middle Cambrian Cadiz Formation of California, which has more prominent relief in lateral profile and is more cap-shaped. In addition, P. valintinei has well defined concentric circles, whereas these are lacking in cf. Discophyllum sp. It also differs from Plectodiscus cortlandensis Caster, 1942 from the Upper Devonian of New York State, as well as other species of Plectodiscus Rauff, 1939 from the Devonian Hunsrück Slate of Germany (Bartels, Briggs & Brassel, 1998; Etter, 2002) and the Carboniferous of Malaysia (Stanley & Yancey, 1986). These have vellelid-like traits, including a sail. They also preserve few radial structures, instead bearing prominent concentric circles that are interpreted as chitinous air canals. Note, regarding the Hunsrück material, here we are referring to the completely preserved specimens illustrated in Bartels, Briggs & Brassel (1998) and Etter (2002). As Bartels, Briggs & Brassel (1998) usefully mentioned, it is not entirely clear if the isolated large disc-shaped structures from this deposit discussed by Yochelson, Stürmer & Stanley (1983) actually represent the same animal; instead these may represent a mollusk, a brachiopod, or salt pseudomorphs (see also Otto, 2000 and references therein). MacGabhann (2012) noted that some specimens of Plectodiscus may represent scratch circles.

Oliver (1984) provided a detailed discussion of Conchopeltis alternata Walcott, 1876 from the Ordovician Trenton Limestone of New York State. Glaessner (1971) and Stanley (1982) treated this species as a chondrophorine (capitate in modern parlance), though Oliver (1984) hesitated to assign it to that suborder. It has prominent radial structures projecting from a circular to ovate interior space; overall, it also has a semi-ovate form. However, it does show some relief in lateral view (perhaps attributable to its preservation in limestone), and some specimens possess four-fold symmetry.

Finally, Caster (1942) considered Palaeoscia floweri Caster, 1942 from the Upper Ordovician of the Cincinnati region to be a porpitid. Such an interpretation is certainly possible. However, specimens are largely devoid of radiating lines except near the central, apical region, where they diverge from a central pore-like structure. Instead, Caster’s (1942) specimens are primarily dominated by prominent concentric bands and thus differ significantly from cf. Discophyllum sp. Again, some specimens of Palaeoscia are almost certainly scratch circles, as is Aysenspriggia Bell, Angeesing & Townsend, 2001, from the Cretaceous of Chile.

Comparisons with miscellaneous fossil medusozoans: Yochelson & Mason (1986) described a specimen from the Mississippian of Kentucky that they cautiously treated as a chondrophorine (capitate of current taxonomy), but its affinities instead seem to belong more likely with the Scyphozoa, as it shows prominent circular coronal muscle bands. This specimen also lacks prominent radial structures. Cherns (1994) described a medusoid from the Late Ordovician or Early Silurian but she suggested it was not a capitate, and we endorse her interpretation. These differ from cf. Discophyllum sp. by the absence of prominent radial structures.

In terms of their relief, the Cararra specimens differ considerably from most species of Scenella Billings, 1872 (e.g., Walcott, 1884; Yochelson & Cid, 1984; Babcock & Robison, 1988; see also discussion in Waggoner & Collins, 1995). Scenella radians Babcock & Robison, 1988 from the Middle Cambrian of Utah does possess lines radiating from the center, KUMIP specimens 204347–204351, but the cap-shaped peak actually hooks slightly backward, which is unlike cf. Discophyllum sp. Further, specimens of Scenella often display much more prominent concentric elements (Yochelson & Cid, 1984). As mentioned in Landing & Narbonne (1992) and Waggoner & Collins (1995), several species of Scenella may in fact be mollusks, and thus the affinities of these would be very distinct from the specimens discussed here.

Comparisons with eldonids: The most apt comparisons for the Carrara specimens seem to lie with several post-Cambrian taxa that have previously been treated as porpitids, but seem instead to have affinities with the eldonids (Conway Morris & Robison, 1988; Dzik, 1991; Conway Morris, 1993; Masiak & Zylinska, 1994; Zhu, Zhao & Chen, 2002; and see MacGabhann, 2012, for a detailed discussion of the eldonids, including a phylogeny). These are characterized by a coiled sac near the center of a discoidal body, representing the digestive tract suspended within a coelomic cavity.

The Carrara specimens are somewhat different from the Cambrian Rotadiscus Zhao & Zhu, 1994, and Pararotadiscus Zhu, Zhao & Chen, 2002, both of which display clear concentric structures and have a dorsal surface which was stiffened. Our specimens also differ from the Cambrian Velumbrella Stasińska, 1960 (previously considered as a porpitid, but which may also be an eldonid), due to the lack of a prominent annulus dividing the inner and outer areas of the disc, and differing style of radial structures; Velumbrella may also have had a stiffened disc surface, as may the potential Ordovician eldonid Seputus MacGabhann & Murray, 2010.

Other eldonids are dominated by radial structures, including internal radial fibers and internal lobes. The Cambrian Eldonia Walcott, 1911, and Stellostomites Sun & Hou, 1987, both display these structures, with post-Cambrian eldonids including Discophyllum Hall, 1847, and Paropsonema Clarke, 1900, displaying radial ridges ornamenting the dorsal surface (MacGabhann, 2012). The radially-arranged features of the Carrara specimens could represent poorly preserved examples of internal lobes or dorsal ornamentation. However, specimens of Eldonia and Stellostomites exhibiting internal lobes universally also preserve the coiled sac even more prominently, with many additional specimens preserving the coiled sac but not the internal lobes (MacGabhann, 2012). It is difficult to envisage how the radial structures in our specimens could represent eldonid internal lobes without also preserving a coiled sac.

However, it may be possible that the radial structures (Figs. 2 and 3) could represent dorsal surface ornamentation. Such ornamentation is seen in post-Cambrian eldonids, including Discophyllum peltatum Hall, 1847, originally described from the Ordovician of New York; Parapsonema cryptophya Clarke, 1900 from the Upper Devonian of New York (see also Ruedemann, 1916); and Paropsonema mirabile Chapman, 1926, from the Silurian of Victoria, Australia. All of these display ridges radiating from a central point, with the coiled sac generally only visible where it is preserved with relief from the surface. It is not inconceivable that the Carrara Formation specimens could be preserving eldonid dorsal surface ornamentation without the relief necessary to highlight the coiled sac.

Both species of Paropsonema show multiple cycles of radial ridges on the surface (MacGabhann, 2012), unlike the specimens described herein. Discophyllum peltatum, however, exhibits only a single cycle of radial ridges extending from the center to the margin. Although the ridges of the Carrara Formation specimens appear to be more irregular that those of Discophyllum peltatum, this could simply be a consequence of a different taphonomic style and poor preservation in the Carrara material. The size and semi-ovate shape of the type material of D. peltatum is also similar to the Carrara discs. A relationship therefore cannot be ruled out, and the Carrara discs are certainly more similar to D. peltatum than any other previously described discoidal fossils.

Due to the lack of clear diagnostic features of D. peltatum in the Carrara material, and the fact that so far only three specimens have been collected from the Carrara Formation, it seems most prudent to refer the Carrara material to cf. Discophyllum sp. The age differences between the material from the Carrara Formation and the Ordovician of New York State may also suggest they are unlikely to represent the same species.

Supplemental Information

Supplemental Information 1 SEM image and spectra of KUMIP 389538

An SEM image and the spectra and weight percentages of elements for the portion of the Discophyllum fossil, KUMIP specimen 389538, in the region demarcated by the blue box labeled “Fig. 4” in Fig. 3A; maps shown in Fig. 4.

Click here for additional data file.

Supplemental Information 2 SEM image and spectra of another part of KUMIP 389538

An SEM image and the spectra and weight percentages of elements for the portion of the Discophyllum fossil, KUMIP specimen 389538, in the region demarcated by the blue box labeled “Fig. 5” in Fig. 3A; maps shown in Fig. 5.

Click here for additional data file.

We thank Paulyn Cartwright (University of Kansas) for discussions about hydrozoan morphology and taxonomy; Jisuo Jin, Brian Pratt, Graham Young, Kenneth De Baets, and an anonymous reviewer for comments on earlier versions of the manuscript; Perry and Maria Damiani for details on locality and site information; and Lisa Amati (NYSM), Bushra Husseini (AMNH), Greg Dietl and Leslie Skibinski (PRI), and Daniel Levin (USNM) for information about the whereabouts of specimens of Discophyllum peltatum.

Additional Information and Declarations

Competing Interests

Author Contributions

Data Availability

The authors declare there are no competing interests. Richie Kurkewicz is an employee of Pangaea Fossils.

Bruce S. Lieberman conceived and designed the experiments, performed the experiments, analyzed the data, contributed reagents/materials/analysis tools, wrote the paper, prepared figures and/or tables, reviewed drafts of the paper.

Richard Kurkewicz contributed reagents/materials/analysis tools, reviewed drafts of the paper.

Heather Shinogle performed the experiments, analyzed the data, contributed reagents/materials/analysis tools, prepared figures and/or tables, reviewed drafts of the paper.

Julien Kimmig analyzed the data, contributed reagents/materials/analysis tools, wrote the paper, prepared figures and/or tables, reviewed drafts of the paper.

Breandán Anraoi MacGabhann analyzed the data, contributed reagents/materials/analysis tools, wrote the paper, reviewed drafts of the paper.

The following information was supplied regarding data availability:

The raw data has been supplied as a Supplementary File.

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
