# Peer review of "Disc-shaped fossils resembling porpitids or eldonids from the early Cambrian (Series 2: Stage 4) of western USA"

_PeerJ, doi:10.7717/peerj.3312_

## Round 0.1 · original submission · Major Revisions

You report interesting, peculiarly preserved discoid structure and extensively investigate their biological origin. I see the merit in describing these fossils and verifying their biological affinity. However, there are still some major points to address before publication:

Title and Structure of the paper: The title is a bit provocative considering the poor documentation and little characters that these specimens possess (see comments by all reviewers). The introduction sets the problems with identifying disc-shaped fossils well. However, it would place the information referring to the specimen rather in the

Material and Methods section. Furthermore, documenting the morphology of the specimens in a more profound before or after their biological origin or affinity would be necessary in the results section at the altest. I therefore suggest to describe and to document the morphology of the specimens in more detail first before scrutinizing the possible biological nature of these structures and discussing their possible affinities. The Title should also be more general – as their identification as Porpitids is far from certain.

Affinities of the specimens: Only a single picture (in rather poor resolution) is provided of your specimens making it hard to recognize their morphology and certainly to verify their relationship with Porpitids; Please provide picture in high resolution of all specimens as well interpretative drawings (see also comments by all reviewers – particularly reviewer 1).

Methods: the used methods for verifying the biological nature of these specimens are sound (verified by all three reviewers), but some additional references might be useful in this context (see comments by reviewer 1). I also agree with reviewer 2 that including all SEM/EDX analysis in the supplementary material should be integrated in the main text as this analysis seems to be main focus of the paper and is crucial for verifying the biological nature of these structures. More clearly showing the delimitation of the fossils would also be useful in this context. It is not really (clearly) recognizable – at least for me where the fossils stop and the matrix starts.

In addition to these and the comments by the reviewers, please also address my comments in the annotated manuscript.

·

Basic reporting

The language is clear, unambiguous and professional.

The introduction and background set the context well. However, there is some information in the introduction regarding the specimens, which I think would be better placed in the Materials and Methods section.

The literature is very well cited, and quite relevant, although I think it would be useful if the authors were to look at some of the work of Schiffbauer and colleagues on the Gaojiashan (e.g. Cai, Y., Schiffbauer, J.D., Hua, H., and Xiao, S., 2012. Preservational modes in the Ediacaran Gaojiashan Lagerstätte: Pyritization, aluminosilicification, and carbonaceous compression. Palaeogeography, Palaeoclimatology, Palaeoecology 326-328, 109-117), and Laflamme and others on the Fermeuse Formation (Laflamme, M., Schiffbauer, J.D., Narbonne, G.M., and Briggs, D.E.G., 2011. Microbial biofilms and the preservation of the Ediacara biota. Lethaia 44, 203-213). These use SEM EDS to examine fossil preservation, and are quite relevant. I also strongly suggest the authors read Petrovich, 2001 (Petrovich, R., 2001. Mechanisms of fossilization of the soft-bodied and lightly armored faunas of the Burgess Shale and of some other classical localities. American Journal of Science 301, 683-726), and my PhD thesis, which could not be more relevant (see also Validity below).

The structure conforms to PeerJ standards modified for the discipline norm.

The figures are relevant and high quality, but I would like to ask the authors if they could provide a sketch interpretation of the fossil, and a sketch of the elemental maps to highlight the area that is part of the fossils, distinguishing it from the area of host sediment.

Raw data is supplied as supplementary information where not in the manuscript, except only one specimen is figured. The authors state that three specimens were collected, all as part and counterpart, but only one is figured; I would very much like to see the other two (even if part of supplemental files).

Experimental design

The manuscript presents original research.

The research question is well defined (what is this fossil, and how was it preserved). However, I would have liked more context on the host formation, and on the surrounding sediment in particular, in terms of mineralogy, sedimentary environment etc.

The manuscript is a rigorous investigation performed to a high technical standard (but again, see also validity).

The Methods are appropriate and are described in sufficient detail to allow replication.

Validity of the findings

The data is robust and sound. The conclusions are well stated, and linked to the question. Unfortunately, I believe they are wrong.

I have a couple of surprises for the authors. One will be a real shock, and welcome. The other, possibly less welcome.

First: I have the type specimens of Discophyllum peltatum. I also have hundreds of new specimens from a different locality on a different continent.

Second: My PhD is in part a complete revision of the group to which Discophyllum belongs (MacGabhann, B.A., 2012. A Solution to Darwin's Dilemma: Differential Taphonomy of Palaeozoic and Ediacaran Non-Mineralised Discoidal Fossils. PhD, National University of Ireland, Galway - available online, https://aran.library.nuigalway.ie/handle/10379/3406). A monograph containing the taxonomy is accepted with the Bulletins of American Palaeontology describing the group systematically. Discophyllum is not a porpitid, but is actually related to Eldonia ludwigi from the Burgess Shale. So are a large number of the taxa mentioned in the manuscript, like Paropsonema, Stellostomites, and Rotadiscus. Pseudodiscophyllum is actually Discophyllum.

Similarly, many of the specimens to which the authors refer are incorrectly interpreted – to no fault of the authors, I hasten to stress. For example, some Plectodiscus and the sole specimen of Aysenspriggia are definitely scratch circles.

Consequentially, I am very sorry, but the entire interpretation section will have to be significantly rewritten.

On the basis of the evidence presented, I do not believe the specimens can be attributed to Discophyllum peltatum.

Additional comments

I am so sorry to do this to you - these are interesting fossils, I am very curious as to their preservation, and you have done a great job reviewing the literature on the unmineralised discoidal fossils. Unfortunately, you assigned them to a group I've spent ten years working on, and I've updated a huge amount of this in an accepted manuscript.

Where to go from here: I would be happy to send you whatever you need from my work in order to rewrite the paper. I am also happy to volunteer to discuss this further with you - just email me at macgabb@edgehill.ac.uk. I can start with photographs of the Discophyllum type specimens.

Reviewer 2 ·

Basic reporting

The manuscript is generally clear and easy to understand, and its literature references provide sufficient background/context. In addition, results of the study are self-contained.

In regards to figures, I have several comments.

First, I would like to see photographs of the other specimens that the authors have identified as Discophyllum. Discs in the fossil record can be cnidarians, benthic organism holdfasts, microbial structures, or various other things! In this case, given the problematic nature of the fossils, additional photos would be very helpful to the reader (images are data!). In addition, currently, the only photograph (Fig. 2) is of a poor quality; its grain-y, pixelated, and supposed ridges are very difficult to see. Given the state of this photograph and that it is the only one available to me, I'm afraid to say that I'm not convinced that the fossils are porpitids as argued. If the ridges ARE there, the photograph is not very convincing. More photographs and those with higher resolution and better lighting/contrast would go a long way in supporting the authors' claims.

Second, given that space is not a big concern in publishing here, the authors should consider including all the SEM/EDS data in their manuscript (it's a bit frustrating to go back and forth between the manuscript and SI).

Third, it would be helpful if the authors included some arrows or boxes to help guide the reader between figures. I would start with the light photography (make it Fig. 1). In it, I would include boxes around the areas studied with SEM and EDS. That way, it's clear exactly what's being shown. Currently, there are too few conspicuous structures in the SEM images to determine where you're looking on the specimen.

Lastly, I highly recommend adding scale bars.

Experimental design

The study describes original primary research, reporting the discovery of new discoidal fossils and their interpretation, and the research question--what are the affinities of these discs?--is well defined and its significance is stated. My main comment regarding the experimental design is that the authors should state more clearly state upfront in the material and methods section their predicitions if there structures were abiotic pseudo- or dubiofossils versus bona fide fossils (i.e. what are the null and alternative hypotheses?). While it is certainly true that SEM and EDS have been used to assess the biogenicity of fossils, the results alone do not prove or disprove biogenicity. For instance, if EDS showed that the structures were pyrite, the structures could have been body fossils or sedimentary structures (diagenetic features). Only by considering the results of SEM/EDS in light of the taphonomy of organisms can we assess the origins of problematic geological features.

Validity of the findings

I believe that this manuscript, as a report of newly discovered discoidal fossils in the Cambrian of the Great Basin, is valid and merits publication. In addition, while I think the hypotheses motivating the SEM/EDS work require some clarification, I concur with the authors' assessment of the discoidal structures as fossils of some type. Could the C, Ca, and P material molding the fossils could be a thin layer of calcium phosphate or some early diagenetic cement, which precipitated as the fossil was decaying in the sediment.

However, beyond these interpretations, this manuscript has many issues. Quite frankly, and with all due respect to the authors, the data (images) do not support the interpretation of the fossils, at least, not to my satisfaction. The authors may convince me, perhaps with inclusion of better photographs or photographs of other specimens. However, all I currently see are discoidal fossils with very low-relief and non-distinct ridges. Neither of these characters (discoidal shape; low-relief ridges) are undeniable features of porpitids. Indeed, the authors mention that the fossils are "scalloped," but despite mentioning various discoidal Ediacara-type fossils (Cyclomedusa, Eoporpita, etc.), they do not discuss arguably the most famous "scalloped" fossil of all time: Aspidella. It is now widely believed that many, though not all, discoidal fossils in the Ediacaran are taphomorphs of Aspidella (i.e. morphological variants of Aspidella varying with regard to their preservational histories). Alhough Aspidella is a trashbin, and may include fossils of various affinities (e.g. jellyfish, microbial structures, sedimentary structures, etc.), the genus sensu stricto likely represents the holdfast of a frond-like organism (indeed, see Lidya Tarhan's recent works on the taphonomy of Aspidella). Taking all of this into consideration, a reasonable alternative explanation for the structures is that they are a taphomorph of Aspidella, and therefore, no necessarily a porportid. Other characters (e.g. tentacles or a sail, as seen in Velella) are needed to substaintiate the interpretation.

I recommend that the authors explore and describe the paleoenvironment of these fossils. The most convincing "jellyfish" fossils are found in mass strandings deposited in tidal and mud flat facies. How does the facies of the Carrara formation containing these discoidal fossils compare?

What is the lithology of the slab containing the fossils? I presume shale because it occurs in the shale member, but Fig. 2 looks kind of coarse, though it may be the image. In any case, I recommend that the authors incorporate additional description of the material.

Additional comments

Overall, I recommend major revisions. In particular, I would present the results as follows...

The authors should introduce the discovery of discoidal structures in the Carrara Formation in the context of the Precambrian-Cambrian transition interval, in which discoidal structures of various types commonly occur. Then, after they have described the material and methods (including a description of their hypotheses for testing with SEM and EDS data), they should present their SEM/EDS results and discuss how those results affirm the biogenicity of the structures. From there, I recommend that the authors present each of the possibilities, that is, the fossils may be microbial structures (see Liu, 2011, Proceedings of the Shropshire Geological Society for examples), holdfasts of benthic possible Ediacara-type organisms, or cnidarians of some type. In an interesting possibility might be that these structures are some sort of holdover from the Ediacaran, perhaps the holdfast of a taxon that surivived well into the Cambrian. Indeed, Conway Morris has described Ediacara-like fronds from Burgess Shale-type localities. Perhaps the discs could be explained by those sorts of taxa. In any case, after even-handedly presenting all the possibilites, the authors can settle on their preferred interpretation of the structures as porpitids. I just want to caution the authors about making any definitive statements about the affinities of the fossils. Speculating that they are porpitids based on their similarities to other putative porpitid fossils is all well and good, but the results are still very equivocal.

·

Basic reporting

This manuscript discusses some small subcircular fossils. While these are potentially of considerable interest, the manuscript is not well structured and the claims made therein are not supported. For those reasons, I must unfortunately recommend that it be rejected in its current form.

It has been famously stated that “extraordinary claims require extraordinary evidence”. To definitely assign small, subcircular, apparently generalized Cambrian fossils to a specific extant group will require preservation of diagnostic specialized morphology, ideally in the form of preserved soft tissues. Evidence of these sorts of features is not presented here.

The structure of the manuscript is a major concern. While there is substantial documentation of the chemical evidence for an organic origin (which is all useful and good), the documentation of the morphology of the fossils is extremely poor. Without a full description and measurement data, it is very difficult for the reader to know what features the fossils actually possess, and this is compounded by the fact that the authors have decided to illustrate them with a single poorly focused photograph (unless I have missed seeing something in the MS). Far better evidence would be needed before I could agree with the authors that they are really onto something, and it is not at all clear to me how they have managed to tentatively assign it to a genus and species.

Experimental design

no comment

Validity of the findings

The assignment of Discophyllum and many other discoidal Paleozoic fossils to porpitids is itself problematic. There is very little published material that has sufficient preserved morphology to make a compelling case for this assignment (the possible exception being the Hunsruck Plectodiscus), and this is made all the more complicated by indications that chondrophores were not present before the Mesozoic (Waggoner and Collins, 2004). Unfortunately resemblance does not demonstrate relationship, and it is likely that many of these fossils could be brachiopods, molluscs, eldoniids, or something else.

This manuscript includes a substantial discussion and comparison with other Paleozoic “porpitids”. If the authors decide to do a revision of the manuscript, I would recommend that they either trim this down to a comparison with those taxa that are truly similar, or augment it so that it is more comprehensive (missing examples would include Chamberlain (1971), Lenz (1980), and several others).

Additional comments

I have a couple of more minor comments:

1. The abstract should probably be written in a far more active style, avoiding phrases like “are discussed” and “is considered.”

2. Figure 2 requires a scale bar – a described magnification is of little use in a paper viewed on screen.

---

## Round 0.2 · Minor Revisions

You addressed all main concerns raised in the previous round of reviews. This was done by describing and illustrating the morphology of the specimens in more detail (including interpretive drawings) as well as integrating the second SEM/EDX analysis in the main text.

I still found some minor points which i would like to see addressed:

Line 1: Title - it might be worth considering maybe having an even shorter title by maybe dropping the Cnidaria: Hydrozoa or even entirely “porpitids (Cnidaria: Hydrozoa)” as you now compare them with eldonids by placing them in Discophyllum. I leave the final decision up to you.

Lines 64-85; i would give the Geology and Paleoenvironment part as an individual subtitle in the introduction (see https://peerj.com/articles/1450/)

Line 119: Do mean “or” or rather “and/or” (could work in synergy) ?

Line 123: I would repeat what kind of trilobite “Bristolia”

Line 258: I guess you mean rather “clear concentric structures”

Line 313-315: or brachiopods - some were even interpreted as salt pseudomorphs: as last discussed by Otto /2000 and references therein) on page 81: Otto M. 2000. Supposed soft tissue preservation in the Hunsrückschiefer (Lower Devonian, Rheinisches Schiefergebirge): the example of brachiopods. Paläontologische Zeitschrift 74:79-89. See also: Otto M. 1994. Zur Frage der “Weichteilerhaltung” im Hunsrückschiefer. Geologica et Palaeontologica 28:45-63.

Line 197-200: I agree with your cautiousness, but if you tentatively assign it to Discophyllum then you kind of say that it is an eldonid. If you are less certain you could also place it as ?Discophyllum sp., but still if you compare it with Discophyllum you kind of say it is probably rather an eldonid than a porpitid.

Line 550: I found out that the Dissertation is available online on various repositories, so please provide a link to one (preferably one accessible to all). As you are often referring to it, I would be easier for readers to follow if they had access to a copy of it.

---

## Author Rebuttal · Round 0.2

March 24, 2017

Dr. Kenneth De Baets
Academic Editor
Dear Kenneth,

Included with this letter is the revised version of our paper (including several new figures). We have tried to address all of the comments and criticisms that you and the reviewers provided (as I shall detail more fully below), and I hope that the paper is now ready for the publication. We thank you for the time and thought that you put into our paper as the academic editor. Also, in case it would be helpful, I have also included the version of our manuscript showing all of the changes tracked. If I can provide any additional information on the changes we have made to our paper please do not hesitate to contact me. Thank you very much for your time and consideration.

Sincerely,

Bruce S. Lieberman
Professor, Department of Ecology & Evolutionary Biology
Senior Curator, Biodiversity Institute
785-864-2741
blieber@ku.edu

How we have addressed the Academic Editor comments and annotations:

We have changed the title of the paper as you requested. We have also provided much more detail on the morphology of the specimens. Further, we have included an interpretative drawing, several new figures, additional references, and integrated the SEM/EDX analysis that was formerly in the supplementary material into the main text. Moreover, regarding the SEM figures, we have delimited the fossils as requested to show where the fossil stops and the matrix starts (as reviewer 2 requested). Indeed, several of the suggestions that you and the other reviewers made involved requests to add or modify figures. The collections manager in the Division of Invertebrate Paleontology at the University of Kansas, Julien Kimmig, has

significant expertise in this area, and he provided yeoman's work in this area, as well as contributions to the text in the manuscript itself, so we included him as an author on the paper.

How we have addressed reviewer 1's comments:

When we had first submitted the manuscript we were not aware of MacGabhann's thesis work, which was very relevant to this paper. Thus, we are grateful to you for selecting him as a reviewer. In his review, he provided very helpful comments and given that he is such an expert on these type of fossils, we recognized that the paper would not be complete without the incorporation of his revised taxonomic concept for the group and the inclusion of his new work in this area. In order for the paper to reach its full potential, we realized that it would make the most sense to ask him to be a co-author and revise the paper along the lines that he suggested in his review, while getting him to incorporate the most up to date knowledge on these enigmatic organisms. Thankfully, he accepted our offer, made the changes he suggested initially, as well as several others, and is now included as a co-author. The changes made in this regard include:

a) Referencing MacGabbhan's thesis at several points throughout the paper, in particular in the discussion of the SEM/EDS work that he had done such as analyses on *Discophyllum* from other localities.
b) Providing a sketch interpretation of the fossil and highlighting where on the specimen the SEM work was conducted so as to distinguish the fossil from the host sediment.
c) We have also provided illustrations of the other specimens as requested.
d) We have corrected the information on the whereabouts of the type specimens.
e) We have adjusted the interpretations of *Plectodiscus* and *Aysenspriggia* in the text.
f) Based on his interpretations and input, we now assign the specimens to the genus *Discophyllum* but no longer include a species level assignment.

How we have addressed reviewer 2's comments:

a) We now provide figures of all of the specimens assigned to *Discophyllum* and have created a new and better figure for the specimen that was originally shown.
b) As requested, we now include the SEM/EDS element maps data in the manuscript (and not as supplementary files).
c) We now also show boxes on the photographs indicating the areas studied with SEM and EDS.
d) We also added scale bars to the figures.
g) We provide more discussion of the affinities of disc shaped fossils, now include a discussion of the affinities of our new material relative to *Aspidella*, and have added references to Lyda Tarhan's work.

How we have addressed reviewer 3's comments:

[Figure]

a) We have added several new figures, and improved the quality of the figures, as requested.
b) We no longer suggest a definitive connection to porpitids and instead suggest that the specimens might represent eldonids or something else.
c) We have provided a scale bar instead of magnification value as requested.

---

## Round 0.3 · accepted · Accept

Thank you for addressing these final suggestions. I noticed one more typo in Otto (2000) - "Hunsrückshiefer" should be replaced with "Hunsrückschiefer", but this can be resolved by the PeerJ staff.

---

## Author Rebuttal · Round 0.3

April 9, 2017

Dr. Kenneth De Baets
Academic Editor
Dear Kenneth,

Thank you very much for your very helpful comments and suggestions from the last round of review of the revised version of our paper. We have made all of the changes you suggested (as I shall detail more fully below), and I hope that the paper is now ready for the publication. Also, in case it would be helpful, I have included the version of our manuscript showing all of the changes tracked. If I can provide any additional information on the changes we have made please do not hesitate to contact me. Thank you very much for your time and consideration.

Sincerely,

Bruce S. Lieberman
Professor, Department of Ecology & Evolutionary Biology
Senior Curator, Biodiversity Institute
785-864-2741
blieber@ku.edu

How we have addressed the Academic Editor comments:

We have changed the title of the paper to "Disc-shaped fossils resembling porpitids or eldonids from the early Cambrian (Series 2: Stage 4) of western U.S.A." as you requested. We have made the "Geology and paleoenvironment" an individual subtitle as you requested. We have also changed the wording to "and/or", repeated that "*Bristolia*" is a trilobite, added the word "clear", and added the mention of brachiopods and salt pseudomorphs and referenced the Otto (2000) paper. Moreover, we have placed the term "cf." in front of *Discophyllum* whenever we reference our material, such that it is now treated as cf. *Discophyllum* sp. Finally, we have added the link in the references section so readers can easily get access to MacGabhann's thesis.